# Brief communication: Implications of outstanding solitons for the occurrence of rogue waves at two additional sites in the North Sea

Ina Teutsch[1], Ralf Weisse[1], and Sander Wahls[2]

[1]Helmholtz-Zentrum Hereon, Geesthacht, Germany
[2]Karlsruhe Institute of Technology, Institute of Industrial Information Technology, Karlsruhe, Germany

**Correspondence:** Ina Teutsch (ina.teutsch@hereon.de)

**Abstract.** We investigate rogue waves in buoy and radar measurement data from shallow depths in the southern North Sea. We analyse the role of solitons for the occurrence of rogue waves by computing discrete soliton spectra using the nonlinear Fourier transform for the Korteweg-de Vries equation with vanishing boundary conditions. In a previous study, data from a single measurement site were considered. The comparison of soliton spectra from time series with and without rogue waves suggested a connection between the shape of the soliton spectrum and the occurrence of rogue waves. In this study, results for two additional sites are reported.

## 1 Introduction

Rogue waves refer to waves that are exceptionally high and/or possess an exceptionally large crest, with respect to the surrounding wave field (Hayer and Andersen, 2000). The surrounding wave field is characterized by its significant wave height, which is the average height of the highest third of waves in a record. In this paper, we consider waves that exceed twice the significant wave height of the surrounding wave field, and/or have a crest height exceeding 1.25 times the significant wave height, to be rogue waves. Rogue waves often appear unexpectedly and thus pose a threat to ships and offshore installations (Bitner-Gregersen and Gramstad, 2016). They can occur in both, deep and shallow water (Didenkulova, 2020). Attempts to predict rogue waves based on sea state parameters have not proven useful for reliable warning systems (Cattrell et al., 2018; Häfner et al., 2021). In our approach, we have transferred the investigation from the time domain to the nonlinear spectral domain, similarly to the procedure in linear spectral analysis. In this study, shallow water is defined as $kh \leq 1.36$, where $k$ is the wave number and $h$ is the water depth. This condition is equivalent to the frequency cut-off for the applicability of the Korteweg-de Vries (KdV) equation used in Osborne (1995). The KdV equation describes weakly nonlinear dispersive unidirectional waves in shallow water with constant depth (Korteweg and De Vries, 1895). The equation can be solved with the inverse scattering method (Gardner et al., 1967). The forward transform in this method can be considered a nonlinear Fourier transform (NLFT) (Ablowitz et al., 1974). It yields a continuous spectrum of oscillatory waves and a discrete spectrum of solitons. The NLFT reveals hidden solitons in a time series that will become visible only later when the series evolve spatially w.r.t. the KdV (Gardner et al., 1967; Ablowitz and Kodama, 1982). In a previous study (Teutsch et al., 2023), we applied the NLFT for the KdV equation with vanishing boundary conditions to time series measured at a specific site in the southern

25   North Sea, and analyzed the resulting soliton spectra. We pointed out that the NLFT was applied as a spectral analysis method, similar to the Fourier transform in the linear case. However, the difference to the linear case is that under ideal conditions, the nonlinear spectrum does not change during wave propagation. Thus, the nonlinear spectral analysis may give an indication on the behavior of the wave train further downstream. The method sheds light on the role that nonlinearity plays in the formation of rogue waves. We found that soliton spectra calculated from time series with rogue waves were more likely to contain a large

30   outstanding soliton. In this study, we extended this analysis to time series measured at additional shallow-water locations in the southern North Sea, and also considered radar next to buoy measurements. Based on these new data, the objective was to determine whether or not rogue wave time series at the additional measurement sites showed the same characteristic signatures in the soliton spectra as derived from the analysis of one individual station in the original analysis.

## 2   Methods

35   ### 2.1   Measurement region and data

We consider half-hour time series from three measurement devices deployed along the German and Dutch coasts in comparably low water depths (Fig. 1). The time series were recorded between 2011 and 2016. At the site WES off the coast of the island Sylt in a water depth of $h = 13\ m$, a Waverider buoy with a measurement frequency of 1.28 Hz was installed, similarly to the site SEE off Norderney that was discussed in Teutsch et al. (2023). At the site AWG in a water depth of $h = 6.3\ m$, a

40   down-looking radar device was mounted to a fixed platform, measuring the air-gap to the sea surface at a frequency of 2 Hz in 2011, and at 4 Hz later in the measurement period.

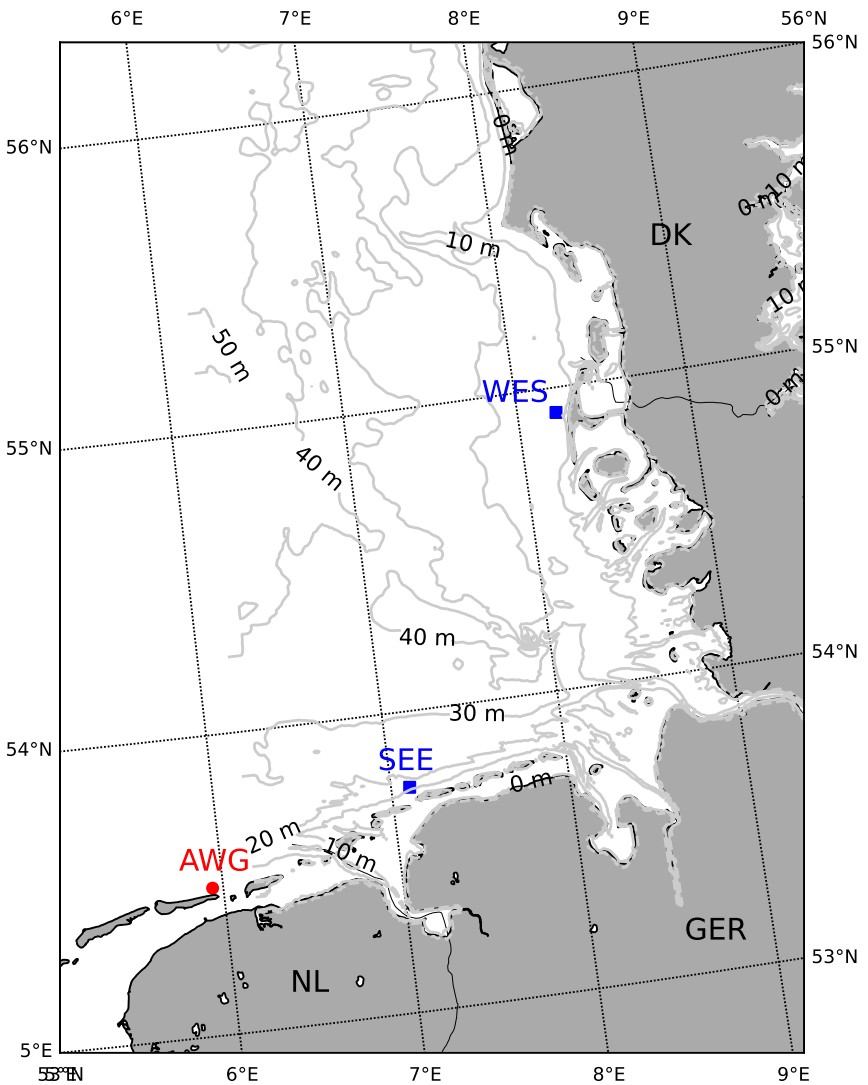

**Figure 1.** Map of the measurement region in the southern North Sea. Blue squares mark the buoy measurement sites, the red dot indicates the location of the radar device. The grey lines refer to isobaths (Dick and Kleine, 2006).

Only samples with large significant wave heights exceeding the local 70th percentile were considered, to exclude measurement uncertainties and to include only large rogue waves relevant for offshore activities. Furthermore, only samples that satisfy

the aforementioned condition of $kh \leq 1.36$ were considered. The data set was split into five categories, based on the largest wave height $H$ and crest height $C$ of individual waves with respect to the significant wave height $H_s$ of the half-hour sample: (i) *non-rogue samples*: $H/H_s < 2$ and $C/H_s < 1.25$; (ii) *height rogue samples*: $2.0 \leq H/H_s < 2.3$ and $C/H_s < 1.25$; (iii) *crest rogue samples*: $H/H_s < 2$ and $C/H_s \geq 1.25$; *double rogue samples*: $2.0 \leq H/H_s < 2.3$ and $C/H_s \geq 1.25$; and (v) *extreme rogue waves*: $H/H_s \geq 2.3$.

## 2.2 Analysis method

The time-like KdV equation Osborne (1993)

$$u_x + \frac{1}{c}u_t + \alpha' u u_t + \beta' u_{ttt} = 0, \; \alpha' = -\frac{3}{2hc}, \; \beta' = -\frac{h^2}{6c^3}, \tag{1}$$

describes the evolution of waves in shallow depths. Here, $u = u(x,t)$ represents the surface elevation at position $x$ and time $t$. The subscripts $_x$ and $_t$ denote partial derivatives in space and time, respectively, Furthermore, $c = \sqrt{gh}$ represents the phase speed in shallow water, where $g$ denotes gravity and $h$ is the water depth. The NLFT for vanishing boundary conditions

$$u(x_0, t) \to 0 \; \text{for} \; t \to \pm\infty, \quad \text{sufficiently fast}, \tag{2}$$

was used to compute the soliton spectra. The soliton spectrum is obtained from the discrete part of the NLFT spectrum, which we calculated using the Matlab (2019) interface to version 0.5.0 of the software library FNFT (Wahls et al., 2018, 2023). This version contains some improvements[1] over the development version used in Teutsch et al. (2023). We reanalysed the data from the station SEE considered in Teutsch et al. (2023), and found only minor differences compared to the results of the previous version. We therefore used the original values from Teutsch et al. (2023) in our discussions of the new results to avoid confusion.

We remark that the choice of the NLFT boundary conditions is not that important in practice for the detection of solitons, e.g. Brühl et al. (2022).

We compared the discrete soliton spectra of time series in the different sample categories. We refer to the spectra as containing an "outstanding soliton" when the ratio $A_2/A_1 \leq 0.8$, where $A_1$ and $A_2$ refer to the amplitude of the largest and the second-largest soliton in the spectrum. A spectrum with a "strongly outstanding soliton" is defined as one where $A_2/A_1 \leq 0.3$.

## 3 Results and Discussion

A comparison between the soliton spectra of time series with and without rogue waves has revealed noticeable differences at the measurement site SEE (Teutsch et al., 2023). Rogue wave time series more often had spectra with exceptionally large outstanding solitons than time series without rogue waves. On the other hand, time series with strongly outstanding solitons very likely contained rogue waves. The measurement data from SEE were part of a larger data set, which was presented in

---

[1]For the vanishing KdV-NFT, the improvements are that spacing of the numerical grid used to localise solitons is now guaranteed to stay below a user-provided parameter, and that the scattering process is normalised to avoid numerical overflow. The grid spacing parameter used in this study was 0.001.

**Table 1.** Share of samples in each category that showed an outstanding soliton, compared to the remaining solitons in the spectrum. Each row represents one of the considered measurement sites.

| Site | Non-rogue | Crest | Height | Double | Extreme | No. of samples |
|------|-----------|-------|--------|--------|---------|----------------|
| SEE  | 36%       | 64%   | 57%    | 72%    | 87%     | 15,156         |
| WES  | 36%       | 64%   | 55%    | 74%    | 76%     | 28,244         |
| AWG  | 22%       | 47%   | 41%    | 60%    | 87%     | 30,214         |

Teutsch et al. (2020). To assess to what extent the previous findings hold at other stations or can be generalized, we here repeated the analysis for the only two other shallow-water stations in the North Sea measurement data set.

At all three sites, the percentages of soliton spectra containing an outstanding soliton for the different categories of samples were determined (Table 1). In principle, similar results were found: outstanding solitons in the discrete spectrum were more typical for time series containing a rogue wave than for time series without rogue waves. The majority of samples with an extreme rogue wave showed an outstanding soliton. The fraction of samples in the data set for which this finding was true differed at the considered stations. The distribution was comparable at the two buoy stations (Table 1). Here, samples containing rogue waves with a particularly high crest showed an outstanding soliton in the discrete spectrum more often than height rogue samples. At the radar station AWG, outstanding solitons in the discrete spectrum were rarer in general, but the difference between non-rogue samples and extreme rogue samples was even more pronounced. At this station, the spectra of height rogue samples contained an outstanding soliton more often than those of crest rogue samples. It seems likely that the differences between buoy and radar sites are caused by the measurement techniques: As described in Teutsch et al. (2020), a high-resolution radar instrument is able to identify more rogue waves than a surface-following wave buoy with a lower sampling frequency. Beyond the aspect of the measurement instrument, potential reasons for the differences seen in Table 1 are the different depths, bathymetries and directional spreads at the respective sites. With a water depth of only 6.3 m, AWG is located in a significantly shallower area than the two buoy stations. In very shallow areas, additional nonlinear phenomena not modeled by the KdV equation play a significant role (Kharif and Pelinovsky, 2003). The interaction of waves with the bathymetry also gives rise to different processes that can encourage rogue wave formation (Prevosto, 1998; Soomere, 2010). The finding of fewer outstanding solitons together with the larger number of rogue waves at AWG thus suggests that additional mechanisms may contribute to the formation of rogue waves at this site. While the KdV equation assumes a constant water depth and unidirectional propagation, this is not true under real conditions in the southern North Sea. All measurement sites are located close to the coast and the sea floor at the considered stations is not flat, but rather inclined to different degrees (Fig. 1). The slope of the sea floor has an influence on the nonlinearity of waves and the occurrence frequency of rogue waves (e.g. Zeng and Trulsen (2012)).

Figure 2 shows more detailed information on the ratio between the largest and second-largest soliton in the discrete spectrum, for one measurement site in each plot. WES and AWG show qualitatively similar results, which are comparable to the result at the site SEE presented in Teutsch et al. (2023). The bulk of non-rogue time series is located above the 80% line (64% at

WES, 78% at AWG), thus not showing an outstanding soliton in the discrete spectrum, while the largest fraction of extreme rogue time series is located below the 80% line, thus, containing an outstanding soliton in the discrete spectrum (76% at WES, 87% at AWG). At station SEE, it was found that the largest part of samples on and below the 30% line, thus, containing a "strongly outstanding soliton", were rogue wave samples (83% of the samples, instead of 7.7% in the full data set). We therefore concluded that a strongly outstanding soliton in the spectrum can be interpreted as a strong indicator of a rogue wave in the corresponding time series. At the sites WES and AWG, although rogue wave samples do not represent the majority in $A_2/A_1 \leq 0.3$, the share of rogue wave samples is still larger than in the full data set (WES: 41% instead of 7.6%; AWG: 21% instead of 6.3%). Thus, also at the additional sites, the presence of a strongly outstanding soliton in the NLFT spectrum indicates that the corresponding time series is several times more likely to contain a rogue wave. The statistical connection between the relation $A_2/A_1$ and the physical occurrence of rogue waves suggests that outstanding solitons typically transform already large individual waves into rogue waves Teutsch et al. (2023). The indicator $A_2/A_1$ is expected to remain approximately constant during wave propagation, at least over shorter distances, which suggests the possibility to detect strongly outstanding solitons before they eventually contribute to the formation of rogue waves. Thus, the indicator $A_2/A_1$ may be a useful addition to future rogue wave warning systems.

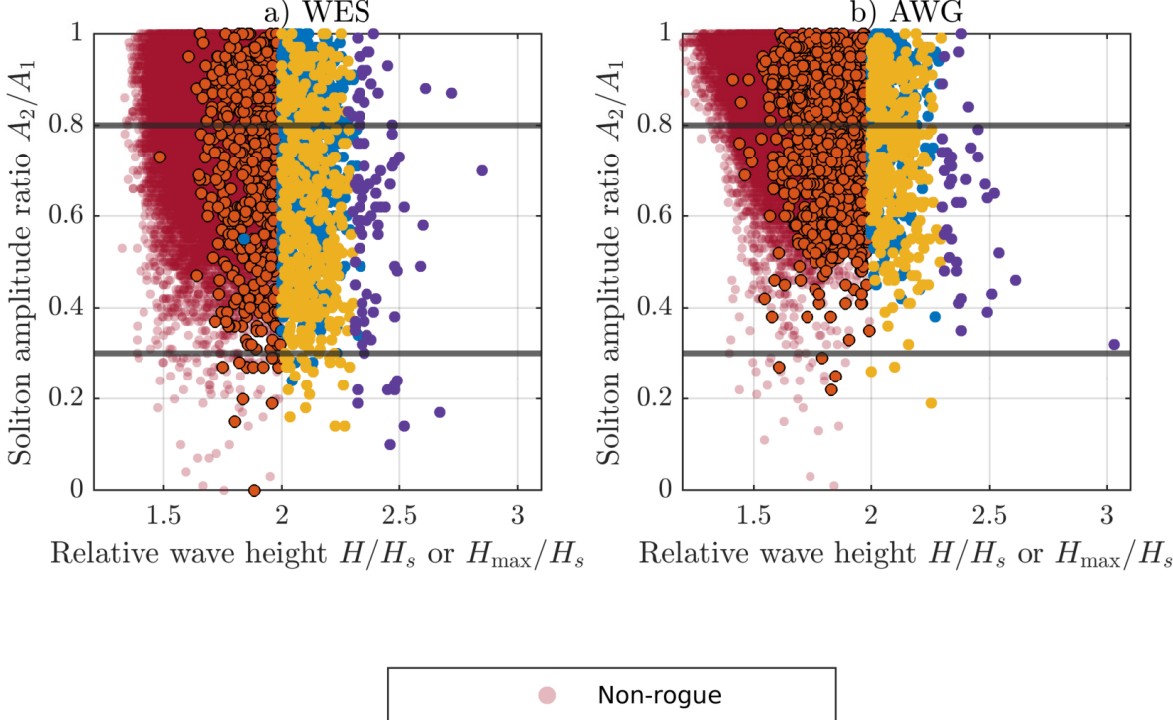

**Figure 2.** Ratio between the second-largest soliton and the largest soliton in the discrete spectrum of measured time series as a function of relative wave height $H/H_s$ or $H_{\max}$ for the different categories of time series. Below the horizontal line denoting 80%, the highest soliton in the spectrum is classified as outstanding. Below the horizontal line denoting 30%, the highest soliton in the spectrum is referred to as strongly outstanding. (a) results from site WES. (b) results from site AWG.

## 4    Conclusions

Based on new data, we confirmed our recent finding that the presence of a large outstanding soliton in the NLFT spectrum is more likely for time series containing a rogue wave than for time series without rogue waves. This was shown at two additional sites in the southern North Sea, of which one was equipped with a different instrument. Therefore, this behaviour does not appear to be site specific (w.r.t. the southern North Sea). On the other hand, the presence of a strongly outstanding soliton in the NLFT spectrum indicated a rogue wave in the corresponding time series, although the indication was not as strong at the additional sites. The detection of strongly outstanding solitons in the nonlinear spectrum could be a useful addition to

rogue wave warning systems. Future research should investigate how well the detected solitons are preserved under realistic propagation conditions.

*Code availability.* The FNFT Software library is available at https://github.com/FastNFT/FNFT/ (last access: 08 September 2023). The version 0.5.0 used for this work is furthermore archived under Wahls et al. (2023).

*Data availability.* The discussed wave buoy and radar data are the property of and were made available by the Lower Saxony Water Management, Coastal Defence and Nature Conservation Agency (buoy SEE), the Federal Maritime and Hydrographic Agency, Germany (buoy WES), and Shell, UK (radar AWG), respectively. They can be obtained upon request from these organizations.

*Author contributions.* All authors contributed to the idea and scope of the paper. IT performed the analyses and wrote the manuscript. RW and SW provided help with data analysis, discussed the results and contributed to writing the paper.

*Competing interests.* The contact author has declared that none of the authors has any competing interests.

*Acknowledgements.* Ina Teutsch received funding for this work from the Federal Maritime and Hydrographic Agency (BSH).

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
