# Peer review of "Brief communication: Implications of outstanding solitons for the occurrence of rogue waves at two additional sites in the North Sea"

_Natural Hazards and Earth System Sciences, 2023_

## Author Comment (AC1)

We thank Anonymous Referee #1 for the constructive comments. In the following, we reply to the individual issues raised by the reviewer. Quotes from the referee's comment are presented in italic, while our answers are printed below each quote in standard text.

**Reply to RC1:**

*"... the measurements by radar seem to demonstrate some noticeable difference, which the authors relate to the higher acquisition frequency of the device. (By the way, the location AWG is characterized by significantly smaller depth. May be, this is the real clue for the observed difference with other locations?)"*

We agree with the reviewer that the differences between buoy and radar measurements can originate from different sources. In our manuscript, we mention the higher acquisition rate of the radar device, as well as different bathymetric conditions at the respective sites and different directional spreading of the wave fields (lines 77-80). Following the suggestion of the reviewer, we will include a comment on the depth conditions at the considered sites and on the influence of a reduced water depth on nonlinear phenomena in a revised version.

*"There is no novelty in either approach or conclusion compared to the preceding paper. […] Thus, the result of this work is just the fact that the work has been done. I cannot recommend publication of this communication."*

Here we disagree with the reviewer's position. We would like to reply that the novelty in our brief communication is the data basis that the investigation has been based on. Our research question was whether the correlations identified in Teutsch et al. (2023) generalize to other measurement sites. We demonstrated that the conclusions obtained from a single station in Teutsch et al. (2023) also hold at other stations and are thus not site specific. This is a novel conclusion and we believe that this is valuable for other researchers and should be published. This is also explicitly mentioned in our reply to RC2. We intentionally chose the form of a brief communication and refer to the journal's guidelines, which state that brief communications "... may be used to [...] disseminate information and data on topical events of [...] interest within the scope of the journal." Following these guidelines, we presented a new data basis which generalizes a previously established connection between the soliton spectrum and the occurrence of rogue wave events from a single station to multiple stations. To address the reviewer's concern, we will further emphasize these points in a revised version.

*"No arguments presented, why the two new locations (but not others) deserve consideration."*

The two sites presented here are the only two remaining sites from Teutsch et al. (2020) that are located in shallow water. The discussed method was established in Teutsch et al. (2023) for a shallow-water site with enhanced rogue wave occurrence. Now the capability of the method is validated for all of the remaining shallow-water stations in Teutsch et al. (2020). We will explicitly mention these arguments in a revised version.

**References**

Teutsch, I., Brühl, M., Weisse, R., & Wahls, S. (2023). Contribution of solitons to enhanced rogue wave occurrence in shallow depths: a case study in the southern North Sea. *Natural Hazards and Earth System Sciences, 23*(6), 2053-2073. doi:10.5194/nhess-23-2053-2023

Teutsch, I., Weisse, R., Moeller, J., & Krueger, O. (2020). A statistical analysis of rogue waves in the southern North Sea. *Natural Hazards and Earth System Sciences, 20*(10), 2665-2680. doi:10.5194/nhess-20-2665-2020

---

## Author Comment (AC2)

We thank Anonymous Referee #2 for the constructive comments. In the following, we reply to the individual issues raised by the reviewer. Quotes from the referee's comment are presented in italic, while our answers are printed below each quote in standard text.

**Reply to RC2:**

*„When this ratio (i.e., the so-called $A_2/A_1$) is small compared to the unity, it will be more likely to occur in time series with a rogue wave. It will be certainly so if the amplitude of the second-largest soliton is larger or similar to the significant wave height of the time series."*

In our analysis, the amplitudes of the solitons in the discrete spectrum were always much lower than the amplitudes of the waves in the corresponding time series. The amplitude of the second-largest soliton $A_2$ was always lower than the significant wave height of the time series. Overall, the soliton train represents only a small fraction of the amplitudes in the time series (see Figure 5a in Teutsch et al. (2023). The situation described by the reviewer did not occur.

*" It is not straightforward to the reviewer why $A_2/A_1$ was chosen for defining an outstanding soliton but not $A_1/H_s$?"*

We thank the reviewer for raising this interesting issue. $A_1$ is indeed related to $H_s$, in that time series with larger waves typically carry larger solitons. We already investigated the behavior of $A_1/H_s$ for the different time series categories at the site SEE in Figure 9 in Teutsch et al. (2023). The normalized soliton amplitudes were higher for rogue than for non-rogue time series, suggesting that solitons play a role in rogue wave generation. However, the difference between the distributions was not as pronounced as the relation between the soliton amplitude ratio $A_2/A_1$ and the time series category. Therefore, we chose the outstanding soliton definition as a rogue wave indicator.

*"Imagine a particular case where both the largest and second-largest soliton within a time series can represent rogue wave events…"*

In all cases the height of the solitons was substantially lower than the significant wave height. The individual solitons alone did not represent rogue wave events. Interaction between the solitons and the oscillatory part of the spectrum is needed to account for such events. We further made sure that each rogue wave time series in our study included exactly one rogue wave. The nonlinear spectra in turn always displayed several solitons. Thus, not each soliton in the spectrum corresponded to a rogue wave in the time series. However, it is indeed possible that a number of solitons contribute to the formation of one rogue wave. This is shown in Figures 6 and 7 of Teutsch et al. (2023). There, we scaled down the rogue wave in a time series and observed several solitons in the nonlinear spectrum to reduce their amplitudes. These solitons were considered to contribute to the amplitude of the considered rogue wave. The largest contribution to the rogue wave is however made by the largest contributing soliton. The parameter $A_2/A_1$ quantifies this contribution.

*"Firstly, the physical meaning of the definition of the outstanding soliton from a stochastic point of view."*

The physical meaning of the outstanding soliton has been discussed in Teutsch el al. (2023). The outstanding solitons are co-located with the rogue waves, but they are much smaller than the rogue waves themselves. The interpretation is thus that an outstanding soliton makes already large waves even larger, pushing them over the rogue wave threshold. There is of course a stochastic element to it. Solitons do not always enhance the amplitudes of other waves, but can also diminish them. (This effect is e.g. nicely illustrated on the cover page of the Phd thesis by Prins (2022), in the second row.) However, given the statistical connection between the indicator $A_2/A_1$ and rogue wave occurrence, we believe that such cases are in the minority. We will repeat the physical interpretation of the outstanding soliton in a revised manuscript.

*"... it would be good to re-think why we need a concept which does not bear a clear physical meaning."*

As pointed out in our previous answer, does the outstanding soliton have a clear physical interpretation. Furthermore, although rogue waves have been studied for decades, warning systems based on sea state parameters are still not performing well. Bearing the need of a warning system in mind, we transferred our investigation from the time dimension to the spectral space, similarly to the way it is done by FFT. The big difference to the linear case is that under ideal conditions, our indicator, which is derived from the nonlinear spectrum, would not change during propagation. That is, a strongly outstanding soliton could be detected before it eventually contributes to the formation of a rogue wave. The true propagation conditions at sea are of course not ideal (i.e., described by the KdV equation), but one can hypothesize that this indicator does not vary too much for meaningful propagation distances. Therefore, we believe that it might be a useful addition to rogue wave warning systems. This is only a hypothesis at the moment that requires more research in the future, but the results presented in this brief communication are supporting this hypothesis.

*"Second, if we have already had a sufficiently long time series, why do we need solitons to relate them to rogue wave events? Because we can directly analyze rogue wave events."*

As already mentioned in the previous response, analyzing the soliton content of time series might be useful for the prediction of rogue waves. The other reason is that they might help us understand the formation of rogue waves better. The identified soliton spectrum characteristics can shed light on the role that nonlinearity plays for the occurrence of rogue waves in shallow water, as opposed to those that are expected within a Gaussian sea state. Again, we emphasize that this is only a hypothesis at the moment that needs further research because it is not yet clear how long the identified soliton components typically persist. However, we point out that for the deep water case, Slunyaev (2021) recently showed that an intense deep-water envelope soliton can persist for a long distance and contributes to the formation of rogue waves. The analysis of measured rogue waves using outstanding solitons, which as our brief communication shows is working at various stations in the real ocean, should therefore only be seen as an intermediate step on the investigation of outstanding solitons for the purpose of rogue wave prediction.

**References**

Prins, P. (2022). Reliable numerical algorithms for the Non-linear Fourier Transform of the KdV equation. *Doctoral Thesis, Delft University of Technology.* doi:10.4233/UUID:171BA94A-E8F4-4969-B6ED-912D4F334968

Slunyaev, A. (2021). Persistence of hydrodynamic envelope solitons: Detection and rogue wave occurrence. *Physics of Fluids, 33*(3), 036606. doi:10.1063/5.0042232

Teutsch, I., Brühl, M., Weisse, R., & Wahls, S. (2023). Contribution of solitons to enhanced rogue wave occurrence in shallow depths: a case study in the southern North Sea. *Natural Hazards and Earth System Sciences, 23*(6), 2053-2073. doi:10.5194/nhess-23